# Investigation and Validation of Numerical Models for Composite Wind Turbine Blades

William Finnegan [1,*] , Yadong Jiang [1] , Nicolas Dumergue [2], Peter Davies [2] and Jamie Goggins [1,*]

[1] Civil Engineering, School of Engineering & MaREI Centre, Ryan Institute, National University of Ireland Galway, H91 HX31 Galway, Ireland; yadong.jiang@nuigalway.ie

[2] Institut Français de Recherche Pour l'Exploitation de la Mer (IFREMER), Centre Bretagne, 29280 Plouzané, France; nicolas.dumergue@ifremer.fr (N.D.); Peter.Davies@ifremer.fr (P.D.)

[*] Correspondence: william.finnegan@nuigalway.ie (W.F.); jamie.goggins@nuigalway.ie (J.G.); Tel.: +353-91-49-2609 (J.G.)

**Abstract:** As the world shifts to using renewable sources of energy, wind energy has been established as one of the leading forms of renewable energy. As the requirement for wind energy increases, so too does the size of the turbines themselves, where the latest turbines are 10 MW with a turbine diameter in excess of 190 m. The design and manufacture of the blades for these turbines will be critical if they are to last for the design life, where the accuracy of the numerical models used in the design process is paramount. Therefore, in this paper, three independent numerical models have been created using three available finite element method packages—ABAQUS, ANSYS, and CalculiX—and the results were compiled. Following this, the accuracy of the models has been evaluated and validated against the results from an experimental testing campaign. In order to complete the study, a 13 m full-scale wind turbine blade has been used, which has been subjected to static testing in both the edgewise and flapwise directions. The results from this testing campaign, along with the blade mass and natural frequencies, have been compared to the results from the independent numerical models. The differences in the models, along with other sources of error, have been discussed, which includes recommendations on the development of accurate numerical models.

**Keywords:** composite materials; experimental test data; finite element analysis; numerical models; offshore wind energy; structural testing; wind turbine blades

## 1. Introduction

Wind energy has now emerged as a leading form of renewable energy, where, at the end of 2019, there was 650.8 GW of wind energy installed worldwide [1]. By the end of 2019, the installed wind energy capacity in Europe had reached 205 GW, where an additional 15.4 GW of new wind power capacity had come online in 2019 [2]. Additionally, across the European Union (EU-28), wind energy accounted for 15% of the electricity consumed in 2019 [2]. As the industry continues to grow, new wind developments are taking place offshore, where the turbines themselves are getting larger as the technology matures. This is evident as the average capacity of wind turbines installed in European waters has doubled, from 2 MW in 2000 to 4 MW in 2014 and SIEMENS Gamesa announced their 10 MW (193 m diameter wind turbine) in 2019 [3]. One of the most critical components of a wind turbine is the turbine blade, which converts the energy of the wind into useful mechanical energy that can be converted into electricity. The design of these blades, both in terms of the aerodynamic shape and structural makeup, is an essential stage of the development of a wind turbine. Therefore, the methods used for the design of the blades must be highly accurate and reliable, and it is essential that the numerical models mimic the real-life blade performance.

The beam element is commonly used for modelling and analysing the structural analysis of wind turbine blades [4–7]. This modelling methodology has a low requirement

for computing resources. However, since the blade is simplified to multiple beam elements, the full-field stress distribution cannot be simulated by this type of element. Hence, the behaviour of composite material under multi-axial stress states cannot be captured by the beam model, resulting in a reduction in accuracy. In recent years, the layered shell element is being more widely used in the structural analysis of wind turbine blades [8–12] and in the design of tidal turbine blades [13–16]. This type of element is suitable for modelling turbine blade since the blade can be considered as a thin-walled structure. As the detailed blade geometry is modelled, the non-linearity introduced by geometry can be captured. Compared to the beam model, this modelling methodology requires more computing resource and has higher accuracy. Moreover, the results given by the shell model contain stress and strain distributions, which can be used in the failure analysis and fatigue life prediction of composite materials. Besides the shell element, the solid element can also be used to model the composite components. Since modelling the composite components using solid elements requires more inputs compared to shell models [17], this modelling methodology is often used in analysing simple composite plates, like the work done by [18–20], but not full-scale wind turbine blades. Peeters et al. [11] analysed and compared the structural behaviour of a 43 m wind turbine blade using both shell element and solid element models. It was found that both shell and solid models predicted results with high accuracy when comparing with the experimental results. However, the solid model had the advantage of simulating stress and strain at the bonding points.

In this paper, three numerical models of composite wind turbine blades are developed and compared. These numerical models have the same input parameters but use 3 different finite element method (FE) packages—ABAQUS, ANSYS, and CalculiX—and were run independently and without prior knowledge of the test results. Test results, which are used to validate the numerical models, are derived from the mechanical (static and dynamic) testing of a full-scale 13 m commercial wind turbine blade. The outputs from the 3 numerical models are compared to the measured results and a discussion on their accuracy, with regard to the predicted mass, natural frequencies, strains, and deflections, is presented, along with a discussion on the potential sources of errors that cause any discrepancies observed in this study.

## 2. Materials and Methods

### 2.1. Aim and Objectives

The overall aim of this study is to investigate the essential parameters for developing the accurate numerical models for composite wind turbine blades that are critical in blade design. In this study, 3 separate numerical models were developed, and experimental testing was performed on the wind turbine blade, in parallel. In order to achieve the aim of the study, a number of objectives must be achieved:

- To determine the relevant input parameters for modelling a composite wind turbine blade,
- To develop 3 separate full-scale numerical models of a composite wind turbine blade,
- To perform experimental physical testing of a composite wind turbine blade, and
- To validate the 3 numerical models by comparing their output to the results from the experimental testing.

### 2.2. Methodology

Initially, the relevant input parameters for a full-scale wind turbine blade, which are the blade geometry, composite design, material properties, and loading, were compiled. In parallel, the full-scale wind turbine blade, which is described in Section 2.3, underwent structural mechanical (static and dynamic) testing and the 3 independent numerical models were developed. These numerical models have exactly the same input parameters but use different FE software packages—ABAQUS, ANSYS, and CalculiX. Numerical predictions on the deflected shape of the blade and strains along the length of the blade were compared to the results from the structural testing in order to validate and contrast the model outputs.

A graphical summary of the process methodology used in this study for fairly comparing the 3 numerical models is presented in Figure 1.

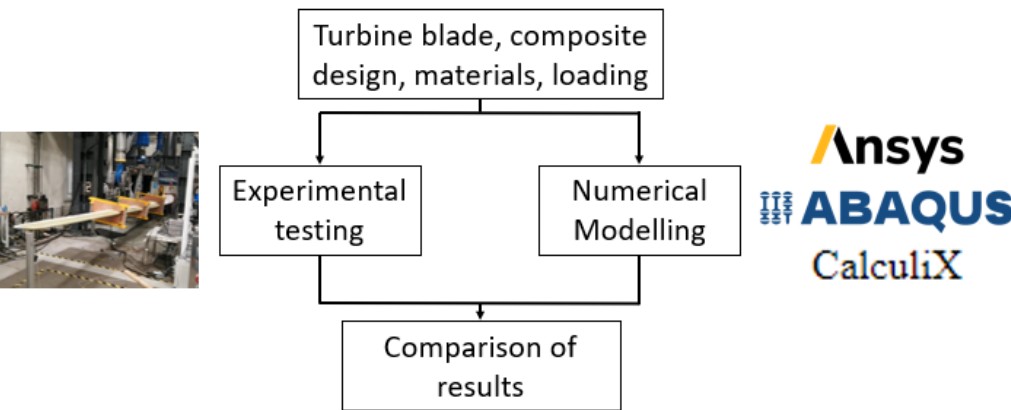

**Figure 1.** Methodology for fairly comparing the 3 numerical models that are developed in this study. Note: A larger version of the experimental testing image is shown in Figure 5.

### 2.3. Wind Turbine Blade Description

The full-scale wind turbine blade tested in this study is a 13 m commercial turbine blade from a 225 kW upwind wind turbine. The blade is 13 m long and its external geometry is constructed with modified NACA 63 series air-foils. A photograph of the blade is shown in Figure 2. The blade is manufactured from glass-fibre reinforced powder epoxy composite material using a novel "one-shot" manufacturing process, which cures the different parts of a wind turbine blade (i.e., skin sections, spar caps web, and root) in one single process to avoid the need for adhesive bonding. Steel inserts in the root of the blade provide a connection to the turbine hub when in operation and to a steel test fixture for the testing campaigns.

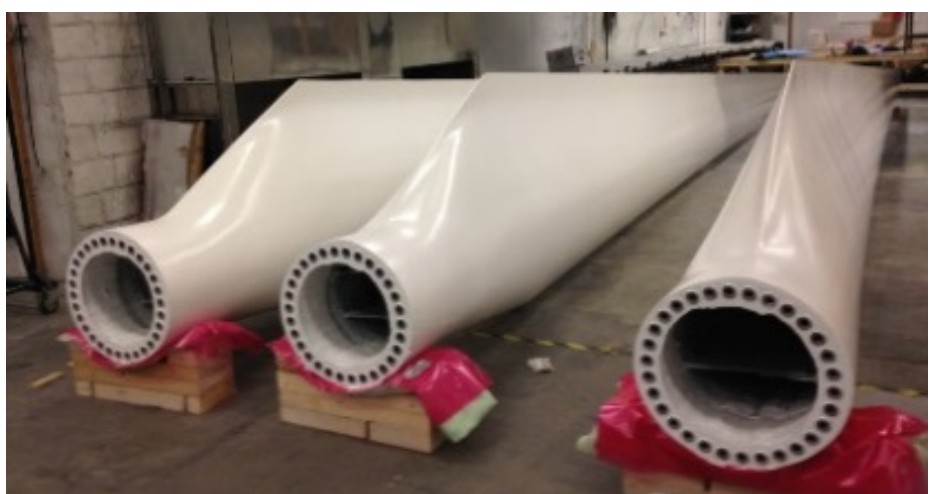

**Figure 2.** 13 m wind turbine blade after being manufactured and finished.

The material properties for unidirectional (UD; 0° direction) and tri-axial (TRI; 0°/±45 directions) fibre orientations are given in Table 1, along with the lightweight polyurethane (PU) core and gelcoat that are used in the wind blade manufacture. The total weight of the blade is 674 kg (including the steel inserts).

**Table 1.** Material properties for the materials used in the current study.

|  | $t$ [mm] | Density [ton/mm$^3$] | $E_1$ [MPa] | $E_2$ [MPa] | $E_3$ [MPa] | $G_{12}$ [MPa] | $G_{13}$ [MPa] | $G_{23}$ [MPa] | $\nu_{12}$ | $\nu_{13}$ | $\nu_{23}$ |
|---|---|---|---|---|---|---|---|---|---|---|---|
| UD | 1.16 | $1.91 \times 10^{-9}$ | 39,700 | 11,900 | 11,900 | 3670 | 3670 | 3670 | 0.2 | 0.2 | 0.2 |
| TRI | 1.1 | $1.91 \times 10^{-9}$ | 21,477 | 13,530 | 12,041 | 9126 | 3670 | 3670 | 0.49 | 0.12 | 0.15 |
| Gelcoat | 0.5 | $1.83 \times 10^{-9}$ | $1 \times 10^{-5}$ | $1 \times 10^{-5}$ | $1 \times 10^{-5}$ | $1 \times 10^{-5}$ | $1 \times 10^{-5}$ | $1 \times 10^{-5}$ | 0.3 | 0.3 | 0.3 |
| PU | 5 | $8.00 \times 10^{-11}$ | 10 | 10 | 10 | 0.2 | 0.2 | 0.2 | 0.3 | 0.3 | 0.3 |

Based on the structural details supplied by the manufacture, the surface of the blade can be divided into three components, namely the spar cap, the leading panel, and the trailing panel, as shown in Figure 3. Inside the blade, there are two webs, going from 0.25 m from the root up to the tip, connecting the pressure side and suction side. The spar cap and the webs form a box-like beam, which works as the main support component of the blade. Figure 4 shows the layer thickness distributions of the four regions along the blade. To accommodate the steel inserts, the spar cap, the leading panel, and the trailing panel share the same layers at the root region. Since the root needs to transfer moments generated from wind loads to the hub, its cross-section was designed to be the thickest part of the blade, with 19 mm thick UD laminate used. This paper primarily focuses on validating that the numerical models adequately capture the global structural response of the blade (e.g., natural frequencies, deflected shape and strain profiles). Thus, only the composite parts of the blade were modelled in this study, where the steel inserts were simplified using a fixed connection at the root, in order to reduce computational effort. It should be noted that due to the existence of the steel inserts, the cross-section thickness at the root is larger than the values shown in the plots. The spar cap is the major component resisting the flapwise wind loads. Therefore, more UD material was used in it compared to that of the leading and trailing panels. The main function of the leading and trailing panels is to form the aerodynamic shape of the blade. Hence, the two components share the same layup details, and less material is used than in the spar cap. However, to avoid the local buckling failure caused by the thin thickness, a PU layer, which works as a non-structural element, was filled in-between the composite layers to increase the shell thickness. It should be noted that there were two additional TRI piles added to the leading panel during manufacturing, aiming to protect the leading edge.

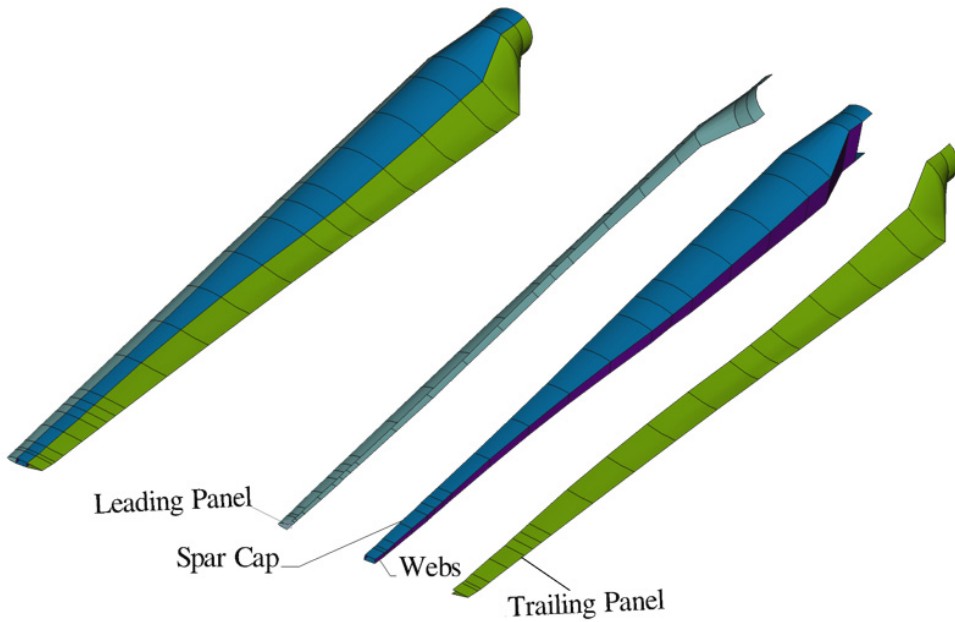

**Figure 3.** Main components of the wind turbine blade.

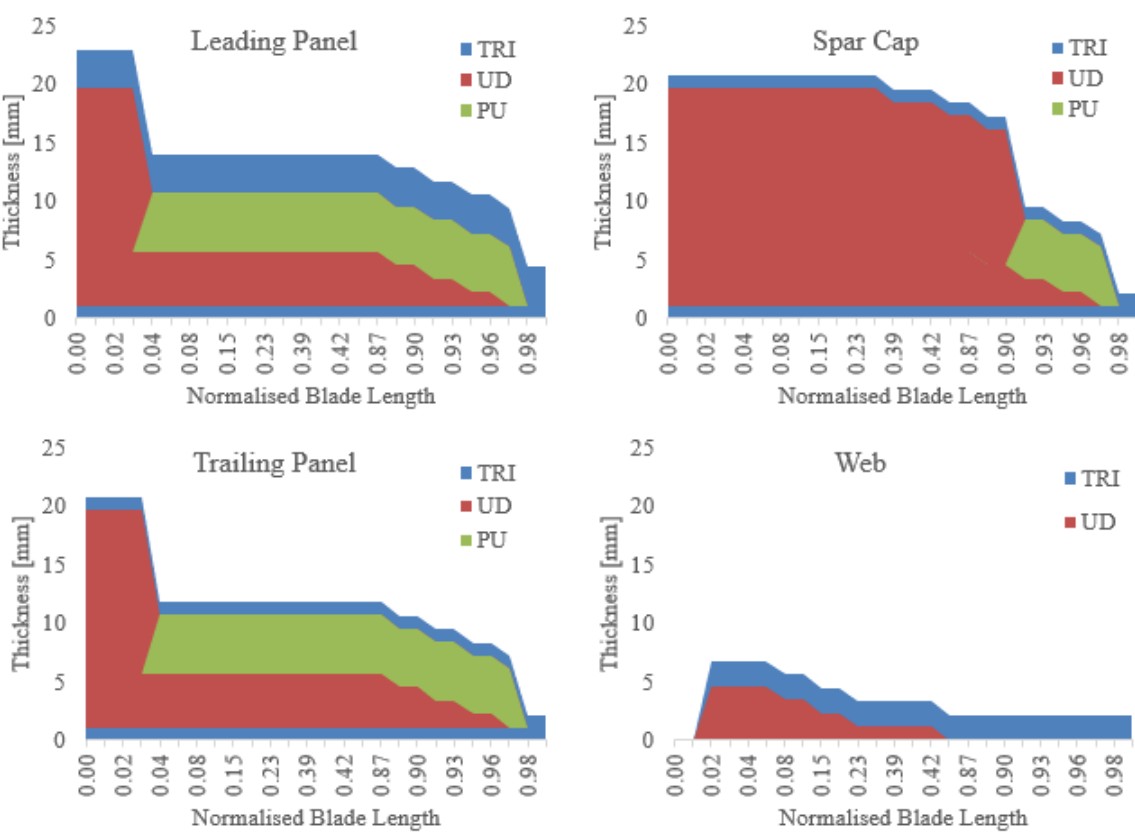

**Figure 4.** Layups details of each components of the wind turbine blade.

## 3. Experimental Testing

The experimental testing of the wind turbine blade was performed using a state-of-the-art multi-actuator load introduction system, which can be seen in Figure 5. The testing campaign was performed at the Large Structures Testing Laboratory at the National University of Ireland Galway in accordance to DNVGL-ST-0376 and IEC 61400-23.

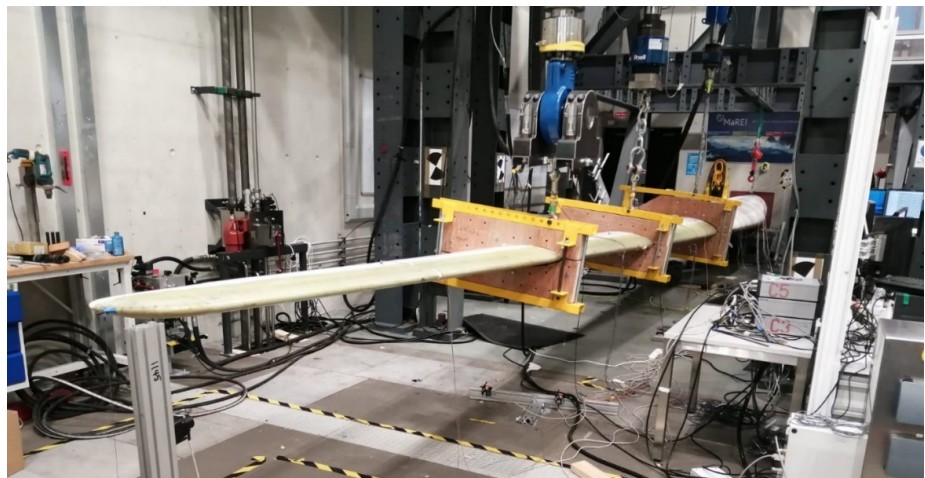

**Figure 5.** Wind turbine blade undergoing static testing in the flapwise direction using a multi-actuator load introduction system.

The temperature in the laboratory was maintained at approximately 19 °C for the duration of the testing programme. The structural testing programme for the blade includes:

- Dynamic testing—to determine the blade natural frequencies, including modal analysis and damping determination.
- Static testing—to determine if the blade can withstand the maximum design load expected during operation.

For the static testing, the wind turbine blade was installed and tested in the flapwise and edgewise directions, respectively. These orientations are defined according to the schematic in Figure 6. Limited by the position of the bolts at the root of the blade, there is a 96° difference between the two orientations in this study. Torsional extreme loads are not considered to be critical for the blade design.

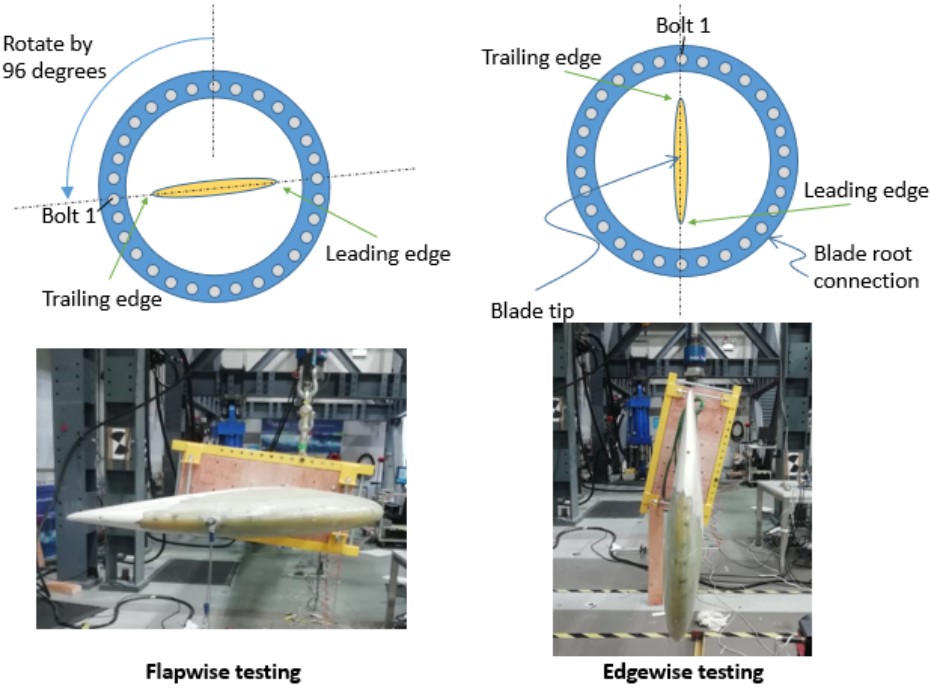

**Figure 6.** Orientation of the installed wind turbine blade during the static testing campaign.

In order to monitor the performance of the wind turbine blade during the testing campaign, comprehensive instrumentation was installed on the blade. The instrumentation used during testing includes:

- Electrical resistance strain gauges applied to the surface of the blade. 6 mm linear strain gauges (with $120 \pm 0.5\%$ $\Omega$ resistance) were used in the spanwise direction along the blade, which have a strain limit of approximately 5%. However, near the root, 6 mm rosette strain gauges were installed on the blade, but only data in the spanwise direction of the blade is presented in this paper.
- Two types of linear displacement transducers, namely the linear variable differential transformer (LVDT) and the string potentiometer.
- Videometric measurements using a 3D laser scanner, a laser scanning vibrometer (LSV), and a GOM digital image correlation (DIC) system.
- Load cells at the locations of load application to the blade.
- Accelerometers, along with the laser scanning vibrometer, for the natural frequency tests.

The output from this instrumentation has been recorded by a National Instruments (NI) PXI data acquisition system and then processed in order to determine the results from the testing, which are used in the comparison in Section 5.

### 3.1. Dynamic Testing

Dynamic testing of the blade was performed in order to determine its natural frequencies. A series of single-axis accelerometers were installed on the suction side of the blade

in strategic locations. Vibration in the blade was excited using an impact hammer to give a transient impact to the blade tip. The natural frequencies and modal properties of the blade are measured before, during and after the static and fatigue testing programmes. These tests were performed without the load introduction fixtures in place. The natural frequencies that were measured are:

- 1st flapwise mode
- 1st edgewise mode
- 1st torsion-wise mode
- 2nd flapwise mode
- 2nd edgewise mode

### 3.2. Static Testing

A static loading was applied to the blade in both the flapwise and edgewise directions in increments of 25% up to the maximum design load, where these load cases have been summarised in Table 2. The design load for the blade has been discretised over the length of the blade at 3 load introduction positions, which gives a good approximation of the bending moment and shear force experienced by the blade during operation. These load introduction positions are at 5.75 m, 8.5 m, and 10.95 m from the root of the blade. Three actuators were employed to distribution the discretised wind loads to the blade. As shown in Figure 5, the point loads from the actuators are transferred to the blade surface through the clamps, which are 100 mm in width around the chord of the blade. The loading direction is always vertically upwards, where the flapwise and edgewise test loads are applied to the blade according to its orientation. Figure 6 defines the blade orientation, in relation to the angle of the blade tip.

**Table 2.** Loads (in kN) applied to the blade at 3 locations during the static testing in both the flapwise and edgewise directions.

|  |  | Load (kN) at 5.75 m | Load (kN) at 8.5 m | Load (kN) at 10.95 m |
|---|---|---|---|---|
| Flapwise | 25% load case | 2.4 | 2.1 | 1.0 |
|  | 50% load case | 4.6 | 3.5 | 2.1 |
|  | 75% load case | 6.6 | 9.9 | 1.8 |
|  | 100% load case | 5.6 | 11.9 | 2.3 |
| Edgewise | 25% load case | 1.3 | 1.1 | 1.0 |
|  | 50% load case | 3.4 | 2.2 | 1.3 |
|  | 75% load case | 4.3 | 3.6 | 1.9 |
|  | 100% load case | 6.3 | 4.8 | 2.4 |

## 4. Model Development

### 4.1. ABAQUS

The static tests were simulated by FE software ABAQUS. The blade geometry was provided by as a STEP file and meshed inside ABAQUS CAE (the pre and post processor of ABAQUS). The blade model was constructed with structural shell element S4. For each shell element, layered shell sections with multiple layers (3 integration points per layer) were assigned. Figure 7 shows the ABAQUS FE model generated for flapwise loading analysis. As can be seen, the load introduction mechanism used in the tests is simulated as rigid links.

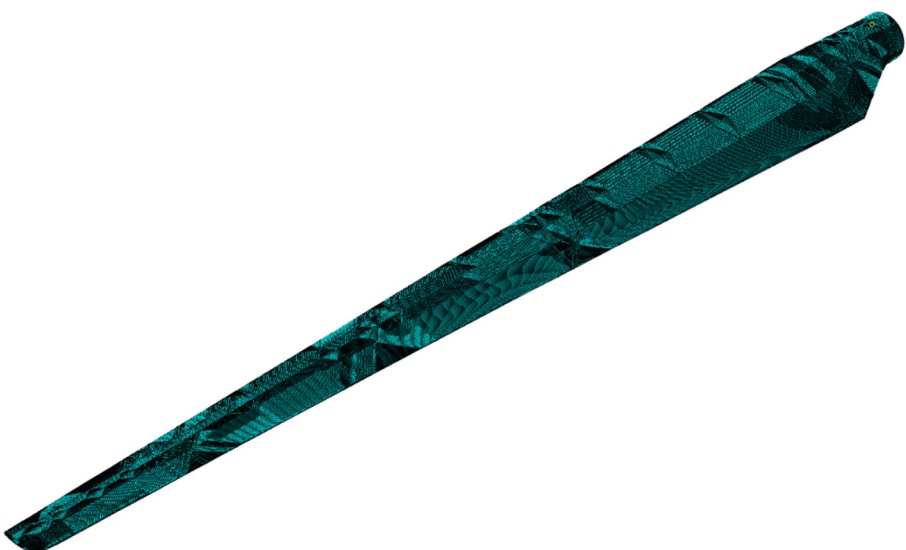

**Figure 7.** The meshed FE model generated in ABAQUS.

*4.2. ANSYS*

The static tests were simulated by ANSYS Mechanical APDL FE software [21]. The blade geometry was generated and meshed by the NUIG in-house developed turbine blade design and optimisation software, BladeComp [22]. Given that the wind turbine blade is a thin-walled structure, the deformed blade can be considered to be in the plane-stress state. Hence, the blade model was constructed with the structural shell element SHELL281, which contains 8 nodes with six degrees of freedom at each node. For each shell element, layered shell sections with multiple layers (3 integration points per layer) are assigned. Figure 8 shows the ANSYS FE model generated for flapwise loading analysis. As can be seen, the load introduction mechanism used in the tests is simulated. This is achieved by utilising the MPC184 rigid link elements to distribute the point loads from the actuators to the blade surface.

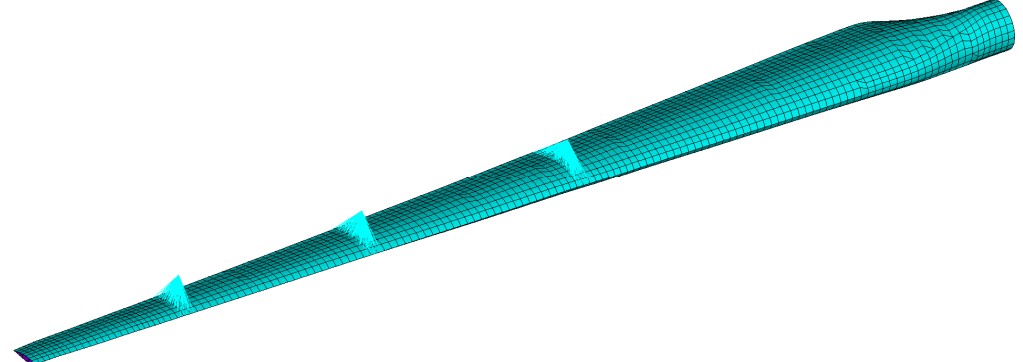

**Figure 8.** The meshed FE model generated in ANSYS Mechanical APDL (for flapwise analysis).

*4.3. CalculiX*

Similar to the ANSYS FE model, the CalculiX [23] FE model was also generated by the NUIG in-house developed software, BladeComp [22]. However, for the element types, the 20-node brick elements C3D20R and the 15-node wedge element (C3D15) were used. For each element, multiple layers with different composite materials are assigned. Similar to the solid elements defined in Peeters et al. [11], one solid element is generated along the normal direction of the shell surface. To achieve an accurate simulation, the integration point number of an element is proportional to the number of assigned composite layers. During analysis, the material properties at each integration point are obtained by

interpolation from the composite layers based on their position. Figure 9 shows the model generated in CalculiX. However, differing from the ANSYS model, the load introduction mechanism is not simulated in the CalculiX model. Instead, the point loads applied by the actuators are uniformly distributed to the blade surface, through the red points highlighted in Figure 9.

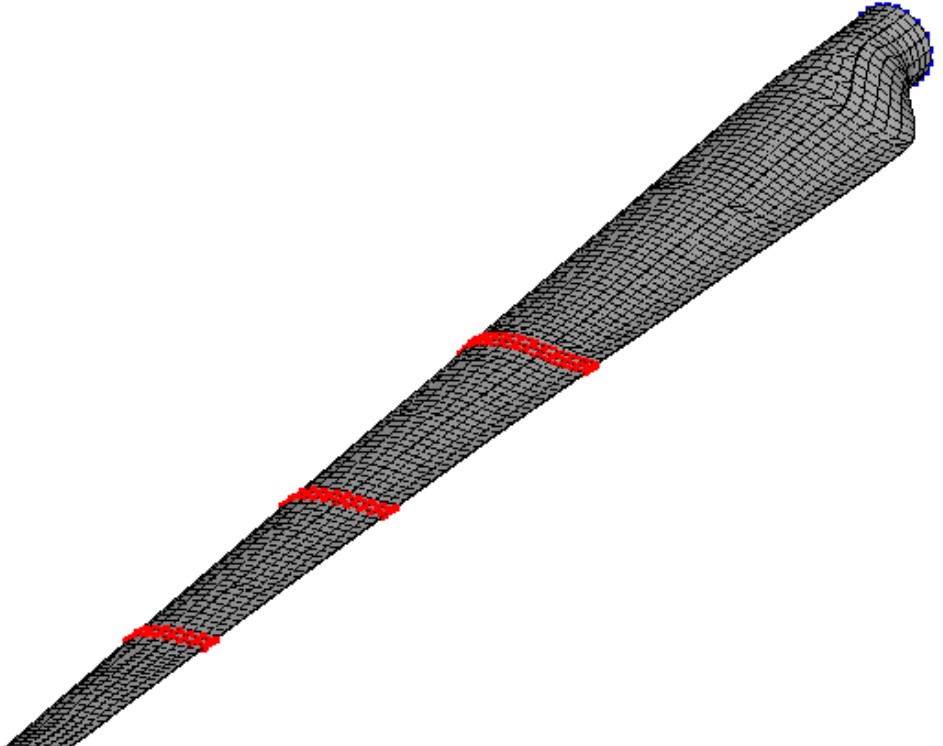

**Figure 9.** The meshed FE model generated in CalculiX.

## 5. Model Validation and Comparison

In order to investigate the accuracy of the 3 numerical models, the outputs from the models are compared to the results from the structural testing. Initially, the blade mass and natural frequencies are compared using the results of the dynamic test. Following this, the deflection of the blade and the strains along the length of the blade are compared using the results of the static test in both the flapwise and edgewise directions.

### 5.1. Dynamic Test

The dynamic test was conducted using a hammer, with accelerometers to measure the resulting accelerations of the blade, as detailed in Section 3.1. The natural frequencies of the blade that are derived from the results of the dynamic test, along with blade mass, are presented in Table 3, where the quoted numbers represent the errors between numerical results and experimental data. These results are compared to the equivalent output from each of the 3 numerical models.

The blade mass, which has been measured using a 6 tonne capacity weighing scales suspended from a gantry crane, is 674 kg. Each of the 3 numerical models have underestimated the blade mass, where the ANSYS model has the largest error of 8.7%. One reason for the models underestimating the mass is that the steel inserts have not been included.

As can be seen in Table 3, all the models can predict the first three natural frequencies of the blade based on the given mode shapes. However, the models start to give mixed local and global deformed shapes when the mode number is higher than 3, which makes it difficult to decipher the exact mode shapes. The numerical models marginally overestimated the flapwise natural frequencies of the blade, where the ANSYS model was

the closest to predicting the measured value. The 1st edgewise natural frequency was reasonably predicted by the 3 numerical models, while the ABAQUS model had a slight under prediction of 4.07 Hz compared to the measured value of 4.25 Hz. The ANSYS model also underestimated (3.62 Hz), while the CalculiX model overestimated the 1st edgewise natural frequency with 4.68 Hz. Only the ABAQUS model was capable of predicting the torsion-wise natural frequency, but it was not accurate, compared to the measured values, as the predicted natural frequency was half that of the measured value. As the steel inserts were not considered in the three models, the additional mass contributed by these components could lead to the differences.

**Table 3.** Comparison of the blade mass and natural frequencies between results from the three numerical models and the experimental testing.

|  | ANSYS | ABAQUS | CalculiX | Experimental |
|---|---|---|---|---|
| Blade mass (kg) | 615.2 (−8.7%) | 652.6 (−3.2%) | 633 (−6.1%) | 674 |
| **Natural Frequency Mode:** | | | | |
| 1st Flapwise (Hz) | 2.74 (7%) | 2.86 (11.7%) | 2.82 (10.2%) | 2.56 |
| 1st Edgewise (Hz) | 3.62 (−14.8%) | 4.07 (−4.2%) | 4.68 (10.1%) | 4.25 |
| 2nd Flapwise (Hz) | 8.34 (7.3%) | 8.61 (10.8%) | 8.70 (12%) | 7.77 |
| 2nd Edgewise (Hz) | - | - | 18.31 (12.9%) | 16.22 |
| 1st Torsion-wise (Hz) | - | 17.65 (−46.9%) | - | 33.27 |

### 5.2. Flapwise Static Test

The load for the flapwise static test was applied to the blade using a multi-actuator load introduction system at 3 point locations on the blade, as detailed in Section 3.2. The deflected shape of the blade and the strains along the centre of the blade spar caps at the outer surface were measured during the structural testing. These results have been compared to the outputs from the 3 numerical models in Figures 10 and 11.

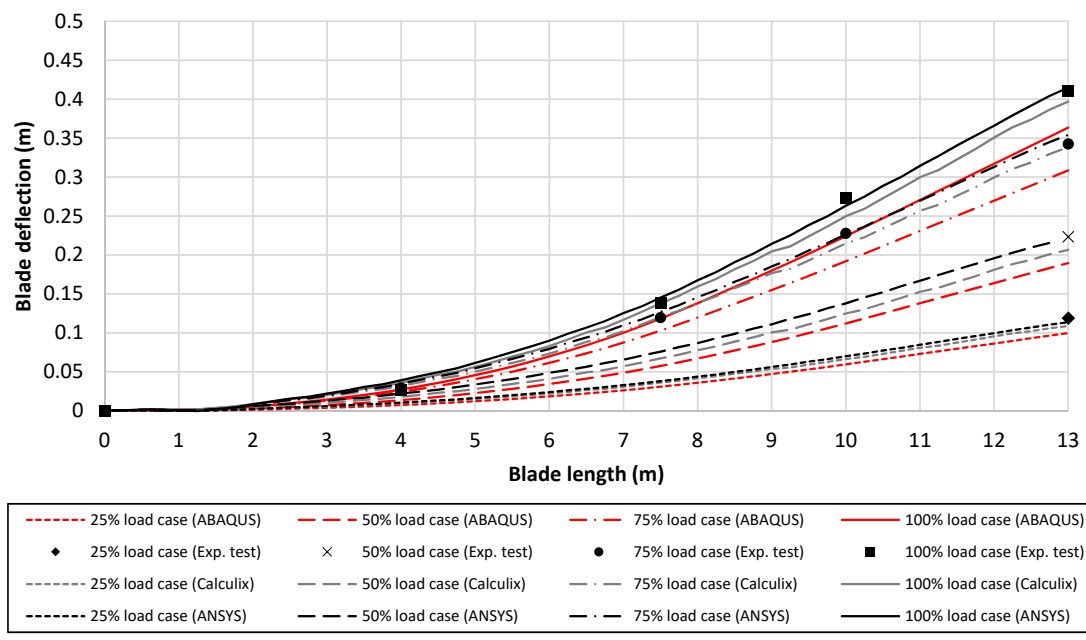

**Figure 10.** Comparison between the results from the 3 numerical models (ABAQUS, ANSYS, and CalculiX) and the results from the experimental static test showing the deflection along the blade (in m) for each of the load cases in the flapwise direction.

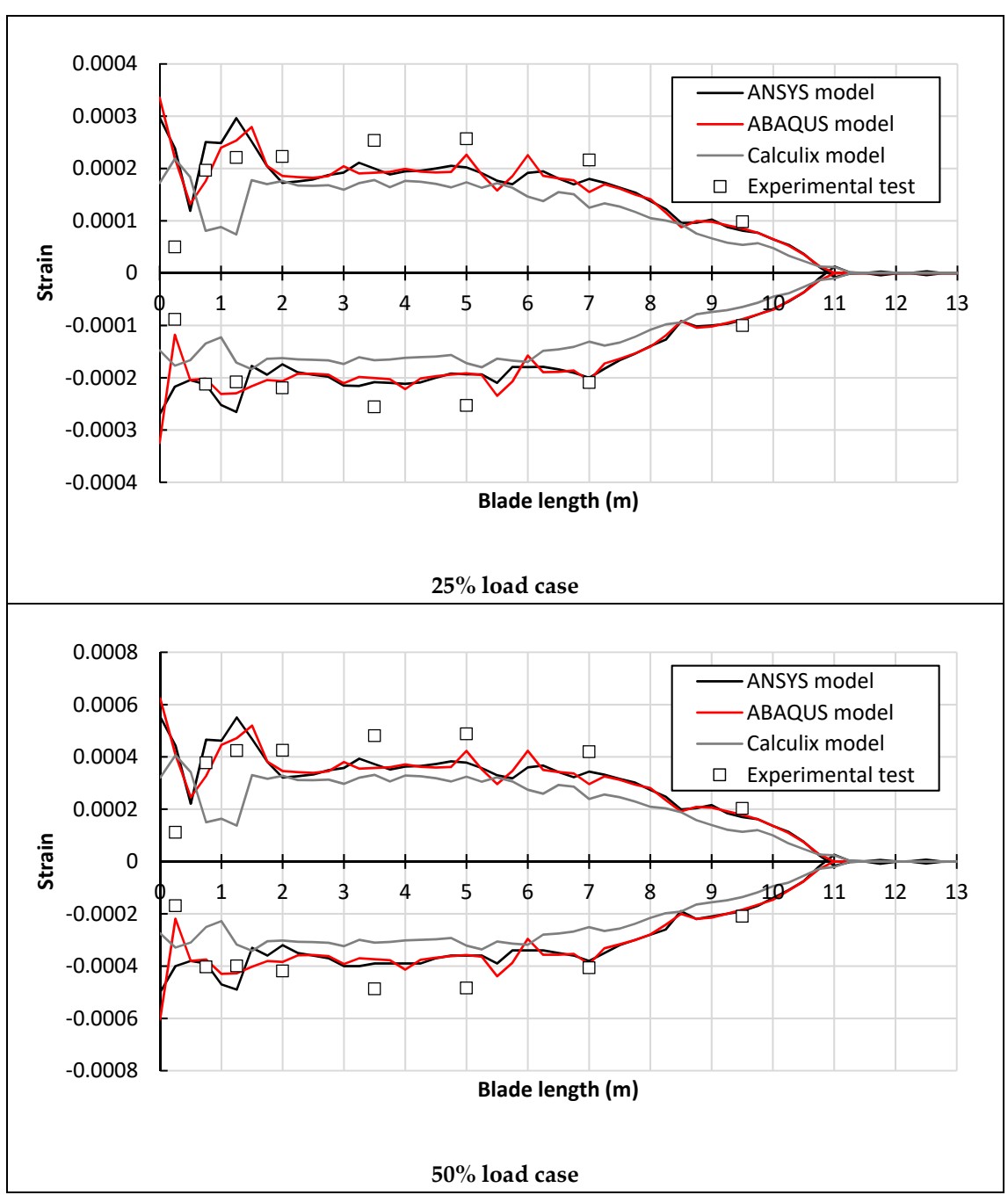

**Figure 10.** *Cont.*

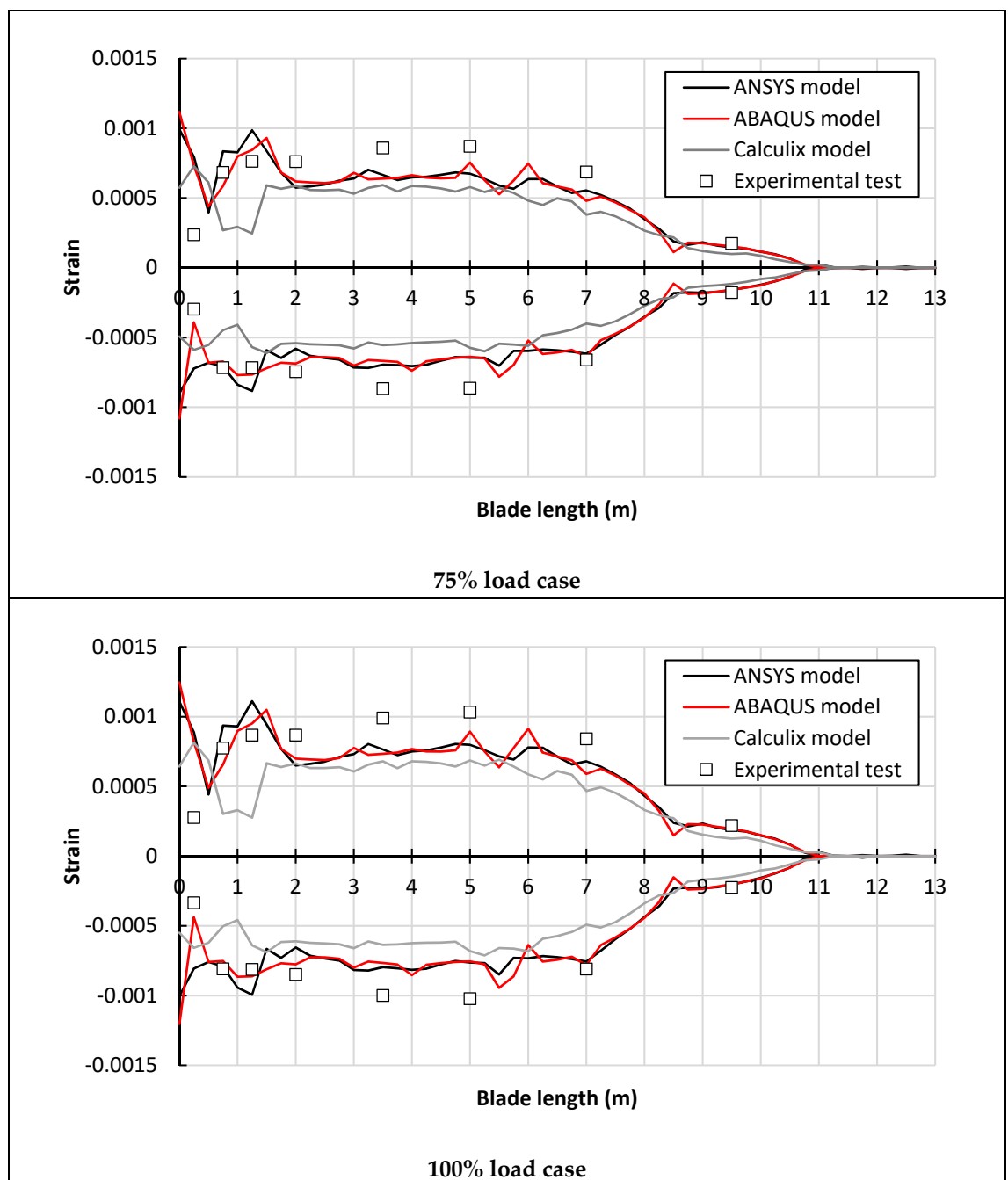

**Figure 11.** Comparison between the results from the 3 numerical models (ABAQUS, ANSYS, and CalculiX) and the results from the experimental static test showing the strain at the center of the spar caps (showing the pressure side (positive strain) and suction side (negative strain)) on the outer surface along the blade length for each of the load cases in the flapwise direction.

The blade deflections are presented in Figure 10, where measurements were taken at 4 locations (at 4 m, 7.5 m, 8 m, and 13 m (blade tip) from the root) during the physical testing. However, it should be noted that only tip deflection data is available for the 25% and 50% load case in the flapwise direction due to issues with stringpot displacement sensors at the other locations. In general, the estimations from the 3 numerical models agree well with the measured values. The ANSYS and CalculiX models are in very good agreement with a difference of 1.1% and −3.3%, respectively, in the deflection at the tip for the 100% load case, compared to the measured value of 0.41 m. The estimate for the tip deflection

from the ABAQUS model is 0.363 m, which is a difference of −11.5% compared to the measured value. However, this is still a reasonable agreement with the results from the structural testing.

The strains along the centre of the blade spar caps at the outer surface were measured during the structural testing and these have been compared to the equivalent output from the 3 numerical models in Figure 11. The strain on the pressure side of the blade is in tension so the strain is positive, while the strain on the suction side is in compression so the strain is negative. From Figure 11, it can be assessed, qualitatively, that the 3 numerical models underestimate the strains along the length of the blade for each of the 4 load cases. The ABAQUS model and ANSYS model predict very similar strains along the length of the blade, which agree well with the measured values from the structural testing near the root (0.75–2 m from the root) and tip of the blade (7–10 m from the root), with reasonable agreement in the inner sections (3–7 m from the root) of the blade. It should be noted that although the blade root section suffers the high moment, the measured strain values at 0.25 m from the root are significantly lower than those at 0.75 m from the root. Moreover, the measured strain values at 0.25 m from the root are also less than that predicted by the three FE models. This may be due to the influence of the steel inserts, which can increase blade strength at root. The FE models did not consider the steel inserts, which can explain these differences. However, the CalculiX model slightly underestimates the strain compared to the other two models but is still in reasonable agreement with the measured values. This may be caused by the element types since the CalculiX FE model was generated from the layered solid element, while the ABAQUS and Ansys models use the shell layered element. Different element types employ different methodologies, which may influence the result accuracy.

### 5.3. Edgewise Static Test

The load for the edgewise static test was imparted on the blade using a multi-actuator load introduction system at 3-point locations on the blade, which is detailed in Section 3.2. The deflected shape of the blade and the strains along the leading edge and trailing edge of the blade at the outer surface were measured during the structural testing. These results have been compared to the outputs from the 3 numerical models in Figures 12 and 13.

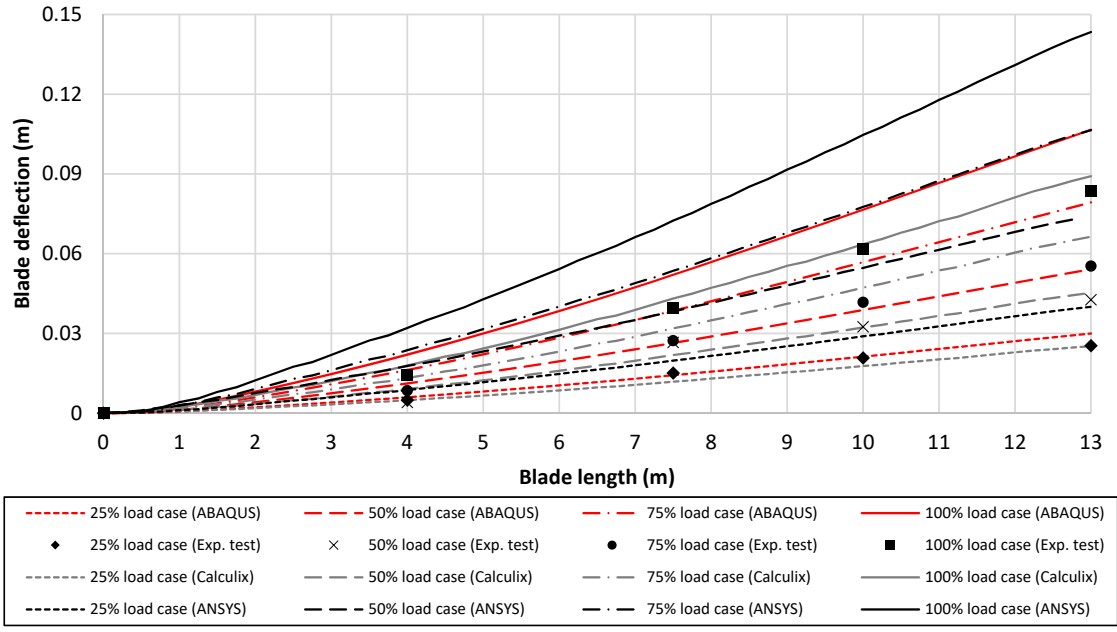

**Figure 12.** Comparison between the results from the 3 numerical models (ABAQUS, ANSYS, and CalculiX) and the results from the experimental static test showing the deflection along the blade (in m) for each of the load cases in the edgewise direction.

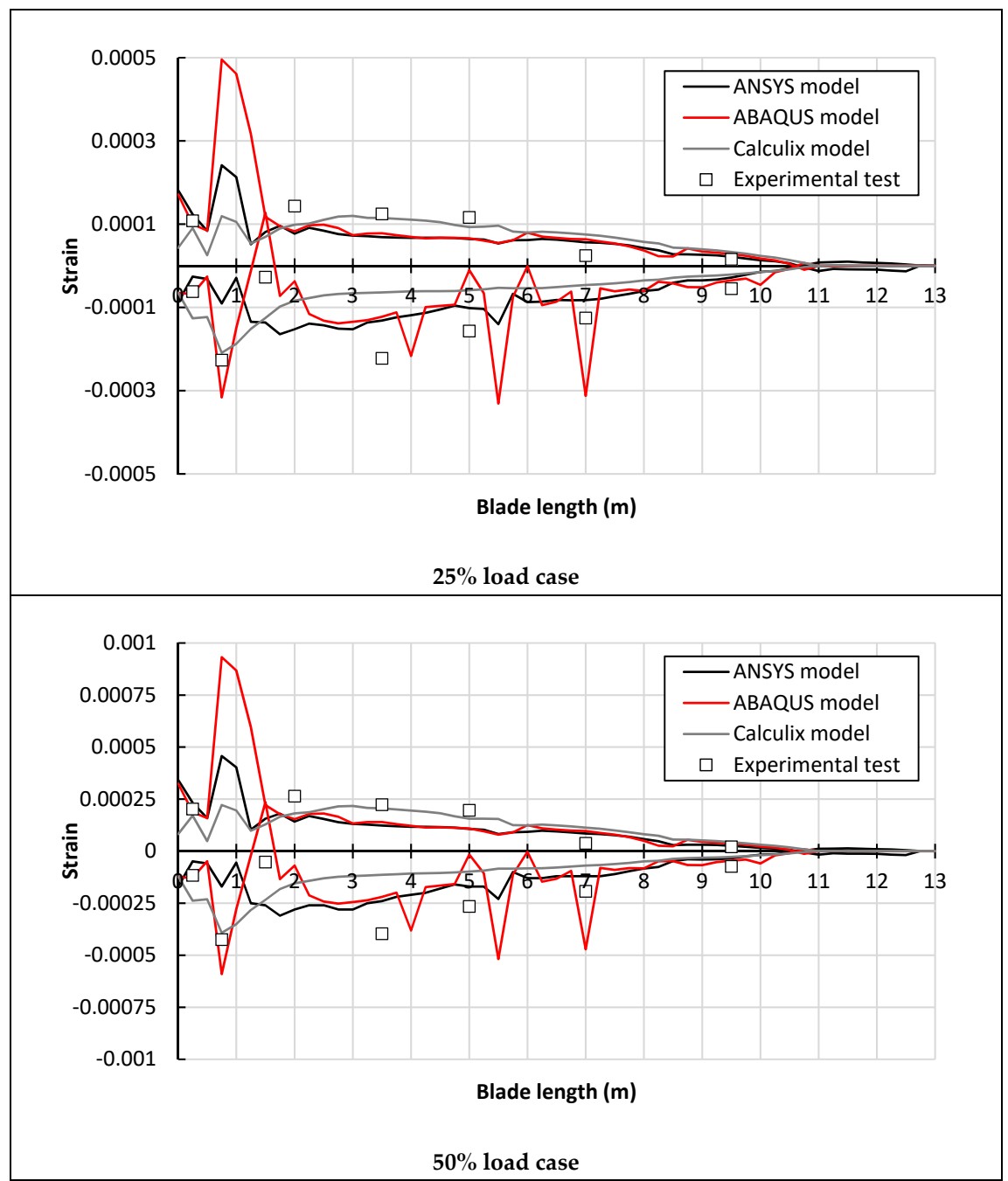

**Figure 12.** *Cont.*

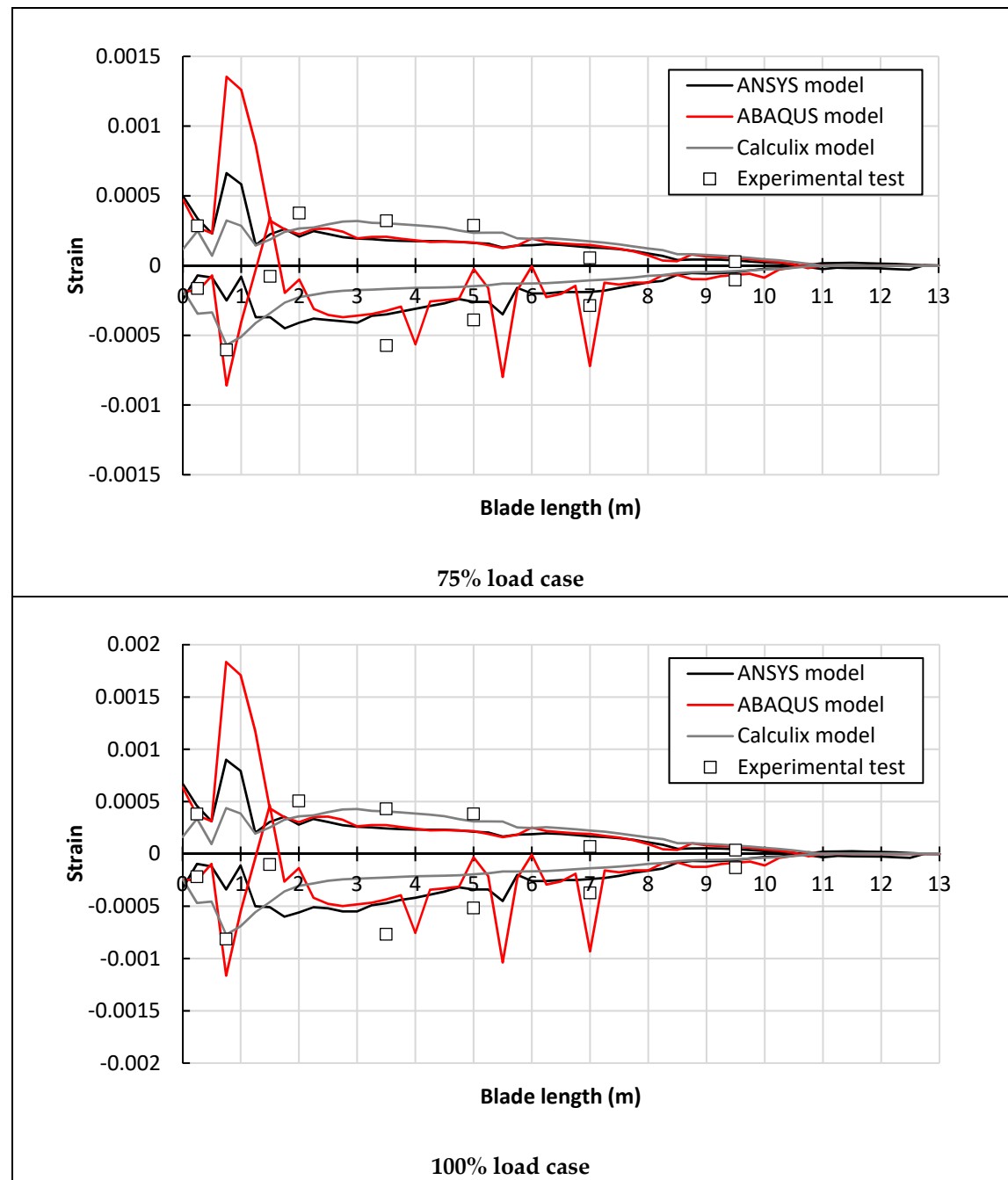

**Figure 13.** Comparison between the results from the 3 numerical models (ABAQUS, ANSYS, and CalculiX) and the results from the experimental static test showing the strain along the leading edge and trailing edge of the blade (showing the leading edge (positive strain) and trailing edge (negative strain)) on the outer surface along the blade length for each of the load cases in the edgewise direction.

Similar to the flapwise static test, blade deflections were recorded at 4 locations (at 4 m, 7.5 m, 8 m, and 13 m (blade tip) from the root) during the physical testing. A summary of these observations, along with a comparison to the output from the 3 numerical models, are presented in Figure 12. All 3 numerical models overestimate the blade deflection, which is evident from Figure 12. However, the CalculiX model is in good agreement, with a difference of 6.6%, in the deflection at the tip for the 100% load case, compared to the measured value of 0.083 m. The ABAQUS model has reasonable agreement with the physical experiment as it estimates the tip deflection for the 100% load case to be

0.107 m, which is 27.5% greater than the measured value. The ANSYS model significantly overestimates the blade deflection, where the tip deflection for the 100% load case is 0.143 m, which is 71.5% greater than the measured value.

The strains along the leading edge and trailing edge of the blade at the outer surface were measured during the structural testing and these have been compared to the equivalent output from the 3 numerical models in Figure 13. The strain on the leading edge of the blade is in tension so the strain is positive, while the strain on the trailing edge is in compression so the strain is negative. From Figure 13, it can be assessed, qualitatively, that the 3 numerical models underestimate the strains along the length of the blade for each of the 4 load cases. Along the leading edge of the blade, the 3 models predict very similar strains, which have good agreement with the measured values from the structural testing near the root and in the tip half of the blade. However, the CalculiX model predictions are in very good agreement with the measured strain values on the leading edge along the length of the blade. There seems to be a lot of variability in the ABAQUS model for the strain along the trailing edge. The ANSYS and CalculiX models underestimate the strain along the trailing edge, except for near the root, where there seems to be reasonable agreement. A contributing reason for this is the difficulty to quantify the strain along the trailing edge as the strain sensors need to be placed on either suction or pressure side at the trailing edge, where they were applied to the pressure side for the experimental testing.

## 6. Discussion

In Section 5, the outputs from the 3 numerical models are compared with the results from the experimental testing programme, in terms of mass, natural frequencies, deflections, and strains, in order to investigate the accuracy of the models. Overall, there was reasonable agreement between the 3 numerical models and the results from the experimental testing programme but there are some differences, which are in part due to the differing methodologies used to develop each of the numerical models; a summary of these differences of the 3 FE models are presented in Table 4. Although each of the numerical models uses the same set of input parameters, the selection of FE modelling methodology, including FE software, element types, and loading introduction mechanism, can cause differences in the numerical results.

**Table 4.** Comparison of the modelling methodologies.

|  | ANSYS | ABAQUS | CalculiX |
|---|---|---|---|
| Element type: | Shell element SHELL281 | Shell element S4 | Solid element C3D20R, C3D15 |
| Element stress/strain components: | 3 | 3 | 6 |
| Load introduction mechanism: | Rigid link | Rigid link | Uniform distributed loads |
| Solver: | ANSYS APDL | ABAQUS | CalculiX |

The blade mass given by the 3 numerical models ranges from 615 kg to 653 kg, with a standard deviation of 16.7 kg, which is less than the actual blade mass of 674 kg. One possible reason for the models underestimating the mass is that the steel inserts have not been included. Regarding the natural frequencies, the ANSYS model has the highest accuracy of the 3, with an average difference of 9.7%. From this study, the CalculiX model is found to be more accurate in predicting the blade tip deflections in both flapwise and edgewise testing scenarios, while the ABAQUS and ANSYS models underestimate the blade edgewise stiffness. Considering that the CalculiX model employs layered solid elements and the other two models utilise the shell element models, it can be concluded that the layered solid element is suitable for analysing the blade response under both flapwise and edgewise loading while the shell element-based blade model may not be



recommended for predicting the edgewise deflection. When examining the strain values overall, the 3 numerical models underestimate the strain in both flapwise and edgewise configurations, compared to the results from the experimental testing programme. The ABAQUS and ANSYS models estimate very similar strain results under the flapwise loading. However, in the edgewise direction, the strain values given by each of the two models are rather different. Unlike with the deflection results, the CalculiX model is consistently giving lower strain values compared to the other two numerical models. The comparisons between the strain values under different testing scenarios indicate that the stress values predicted by the 3 numerical models may be underestimated. Considering that the composite laminate failure prediction under extreme loads and the wind turbine blade service life calculation rely on the stress and strain values given by the FE analysis, it appears that the selection of modelling methodology can be a source of uncertainties in the wind turbine design.

## 7. Conclusions

This paper presents a methodology for developing a FE numerical model of a composite wind turbine blade, which was validated against the results from an experimental testing programme. In order to investigate the effect of input parameters and methodologies used in creating the numerical model, 3 different FE software packages were used, independently and without prior knowledge of the test results, to create 3 numerical models of the blade. Overall, there was reasonable agreement between the 3 numerical models and the measured values. There were some minor discrepancies, which are primarily based on the FE modelling methodology used. For predicting the blade response under both flapwise and edgewise loading, models based on 'layered solid elements' and using a 'uniformly distributed load' to model the loading on the blade were found to give a higher accuracy in the FE prediction compared to the results from the experimental testing programme.

The results of this study can be used to increase the accuracy of the numerical models for wind blades, both onshore and offshore, in order to increase confidence in the methodologies used. Resource usage is becoming more of a concern for wind energy developers as they strive to make longer, stiffer blades, with minimal composite material usage, in as sustainable a manner as possible. Highly accurate numerical modelling of blades will enable these aims to be achieved as the world shifts to a greater reliance on clean, sustainable, renewable energy. The results of this study can also be used in other sectors, where the accuracy of the FE numerical models is essential to efficient development, for example the tidal energy and automotive industries.

**Author Contributions:** Conceptualization, W.F., P.D. and J.G.; data curation, W.F.; investigation, W.F., Y.J. and N.D.; methodology, W.F., Y.J. and N.D.; formal analysis, W.F., Y.J. and N.D.; writing—original draft preparation, W.F., Y.J., N.D., P.D. and J.G.; supervision, P.D. and J.G.; funding acquisition, P.D. and J.G. All authors have read and agreed to the published version of the manuscript.

**Funding:** This research was funded by European Commission through the H2020 MaRINET2 project (grant agreement no.: 26255). The first, second and last authors would like to acknowledge the support from Science Foundation Ireland (SFI), through the MaREI Research Centre for Energy, Climate and Marine (Grant no. 12/RC/2302_2) and funding from the European Commission through the H2020 CRIMSON project (grant agreement no.: 971209). The second author would also like to acknowledge the support from SFI for an Industry Fellowship (Grant no. 19/IFA/7417).

**Informed Consent Statement:** Not applicable.

**Data Availability Statement:** Publicly available testing data available in this publication. Additional testing data is available from [24].

**Conflicts of Interest:** The authors declare no conflict of interest.

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
