# Peer review of "Investigation and Validation of Numerical Models for Composite Wind Turbine Blades"

_jmse, doi:10.3390/jmse9050525_

Round 1

Reviewer 1 Report

The paper addresses then experimental and numerical assesment of a 13 meters long wind turbine blade. In general the work is well presented and may intersted a wide scientific community.

Some shortcommings should be clarified before publishing:

1- it is not clear why the steel inserts are not included in the model, considering the possible impact on the results

2-clarify whether rosette or unidirectional strain gauges are used

3- is torsion avoided in testing? How?

4- how is the self weight treated in the comparison between test and numerical? what is the relationship with any nonlinear nature of load-displacement curve?

5- when assessing natural frequencies how is the blade mass corrected in the comparison between test and numerical?

6- in Figure 10 experimental values for 25%, 50% load not shown

7- Conclusion chapter is too general; clear recommendations on the development of accurate numerical models should be highlighted

Author Response

Dear Reviewer,

Thank you very much for taking the time to review our paper. We have edited our manuscript in line with all of the reviewer comments. The following are my responses to your queries (in red):

1- it is not clear why the steel inserts are not included in the model, considering the possible impact on the results

A clarification has now been added to Section 2.3, as follows: “This paper primarily focuses on validating that the numerical models adequately capture the global structural response of the blade (e.g. natural frequencies, deflected shape and strain profiles). Thus, only the composite parts of the blade were modelled in this study, where the steel inserts were simplified using a fixed connection at the root, in order to reduce computational effort.”

2-clarify whether rosette or unidirectional strain gauges are used

Clarifications have been added in Section 3, as follows: “Electrical resistance strain gauges applied to the surface of the blade. 6 mm linear strain gauges (with 120±0.5% Ω resistance) were used in the spanwise direction along the blade, which have a strain limit of approximately 5%. However, near the root, 6 mm rosette strain gauges were installed on the blade, but only data in the spanwise direction of the blade is presented in this paper.”

3- is torsion avoided in testing? How?

Torsion was not avoided in the testing. However, only minor torsion effects were observed in the flapwise direction.

4- how is the self weight treated in the comparison between test and numerical? what is the relationship with any nonlinear nature of load-displacement curve?

The overall weight of the blade was measured in the laboratory using a scales from the gantry crane and this was compared to the calculated output from the 3 models, which have been presented in the paper in Table 3. The blade self weight was tared at the start of each experimental test and the self weight was ignored in the 3 models.

5- when assessing natural frequencies how is the blade mass corrected in the comparison between test and numerical?

As the results are from blind testing/model development, the outputs from the models have been compared directly to those recorded during testing with no correcting for the differing blade weights.

6- in Figure 10 experimental values for 25%, 50% load not shown

Unfortunately, there were issues with the stringpot displacement sensors for these two load cases in the flapwise direction so no data was recorded. I have included the following note in the text to clarify: “However, it should be noted that only tip deflection data is available for the 25% and 50% load case in the flapwise direction due to issues with stringpot displacement sensors at the other locations. In general, the estimations from the 3 numerical models agree well with the measured.”

7- Conclusion chapter is too general; clear recommendations on the development of accurate numerical models should be highlighted

I’ve added the following detail to the conclusion to give some clear recommendations for FE modelling: “There were some minor discrepancies, which are primarily based on the FE modelling methodology used. For predicting the blade response under both flapwise and edgewise loading, models based on ‘layered solid elements’ and using a ‘uniformly distributed load’ to model the loading on the blade were found to give a higher accuracy in the FE prediction compared to the results from the experimental testing programme.”

Yours faithfully,

Dr William Finnegan

Reviewer 2 Report

I liked the manuscript. I understand what the authors did, why, and how. The explanation is clear and logical. I found no objections to publication.

Author Response

Dear Reviewer,

Thank you very much for taking the time to review our paper.

Yours faithfully,

Dr William Finnegan

Reviewer 3 Report

The article deals with the investigation and validation of three numerical models of composite wind turbine blades compared with the experimental measurement. The article is well written, the presentation is good and the results are clear. 

I have only minor remarks. 

The quality of Fig. 1 is low, (the photo of the experimental testing site is hard to see).

In general, the authors should use the vector graphic to present the graphs and diagrams.

All units have to be spaced (e.g. 12 m)!

On page 8 line 237 there is a wrong citation [website].

Section III. Could you please provide more details about the used sensors and described the signal processing of the measured data??

Author Response

Dear Reviewer,

Thank you very much for taking the time to review our paper. We have edited our manuscript in line with all of the reviewer comments. The following are my responses to your queries (in red):

The quality of Fig. 1 is low, (the photo of the experimental testing site is hard to see).

This image is also in Figure 5 so I have redirected the reader with the following note in the caption: “Note: A larger version of the experimental testing image is shown in Figure 5.”

In general, the authors should use the vector graphic to present the graphs and diagrams.

The authors have opted not to use vector graphics.

All units have to be spaced (e.g. 12 m)!

All units have been changed throughout the paper to satisfy this comment

On page 8 line 237 there is a wrong citation [website].

This citation has been corrected.

Section III. Could you please provide more details about the used sensors and described the signal processing of the measured data??

A clarification has now been added to Section 2.3, as follows: “This paper primarily focuses on validating that the numerical models adequately capture the global structural response of the blade (e.g. natural frequencies, deflected shape and strain profiles). Thus, only the composite parts of the blade were modelled in this study, where the steel inserts were simplified using a fixed connection at the root, in order to reduce computational effort.”

The data was recorded by a National Instruments (NI) PXI data acquisition system, which has now been clarified.

Yours faithfully,

Dr William Finnegan